# Extracellular Vesicles as Mediators and Potential Targets in Combating Cancer Drug Resistance

**DOI:** 10.3390/molecules30030498

**Published:** 2025-01-23

**Authors:** Haodong Zhang, Bohan Wu, Yanheng Wang, Huamao Du, Liaoqiong Fang

**Affiliations:** 1College of Sericulture, Textile and Biomass Sciences, Southwest University, Chongqing 400715, China; zhanghaodong1258@outlook.com (H.Z.); duhmao@swu.edu.cn (H.D.); 2Westa College, Southwest University, Chongqing 400715, China; jane721015@outlook.com (B.W.); wyh7786@outlook.com (Y.W.); 3National Engineering Research Center of Ultrasound Medicine, Chongqing 401121, China

**Keywords:** extracellular vesicles, cancer drug resistance, molecular mechanisms, tumor microenvironment, EVs biogenesis, therapeutic interventions

## Abstract

Extracellular vesicles (EVs) are key mediators in the communication between cancer cells and their microenvironment, significantly influencing drug resistance. This review provides a comprehensive analysis of the roles of EVs in promoting drug resistance through mechanisms such as drug efflux, apoptosis resistance, autophagy imbalance, and tumor microenvironment modulation. Despite extensive research, details of EVs biogenesis, cargo selection, and specific pathways in EVs-mediated drug resistance are not fully understood. This review critically examines recent advancements, highlighting key studies that elucidate the molecular mechanisms of EVs functions. Additionally, innovative therapeutic strategies targeting EVs are explored, including inhibiting EVs biogenesis, engineering EVs for drug delivery, and identifying resistance-inhibiting molecules within EVs. By integrating insights from primary research and proposing new directions for future studies, this review aims to advance the understanding of EVs in cancer biology and foster effective interventions to mitigate drug resistance in cancer therapy.

## 1. Introduction

According to WHO statistics, the global incidence of new cancer cases is projected to reach 35 million by 2050 [1], underscoring the critical challenge of cancer control. Drug therapies have provided sustained benefits for cancer control. Unfortunately, almost all cancer drugs can induce resistance, rendering them ineffective against cancer cells and exacerbating the progression of cancer, possibly resulting in patient mortality. The mechanisms underlying resistance to clinical cancer drugs are heterogeneous and multifactorial [2]. Extracellular vesicles (EVs), as mediators in the communication between cancer cells and their microenvironment, dynamically regulate the survival and signaling pathways of cells through various mechanisms [3,4], which leads to a resistance to multiple cancer drugs [5]. EVs are generally classified into large EVs (diameter > 200 nm) and small EVs (sEVs, diameter < 200 nm). Exosomes, a subset of sEVs, originate from the endosomal system and are released through the multivesicular body (MVB) pathway, whereas microvesicles, derived from the plasma membrane, typically refer to larger EVs or all EVs [6,7]. EVs regulate drug resistance by modifying secretion levels and cargo composition [3,4]. The increased secretion and transfer of EVs are indicative of the development and spread of drug resistance. Studies have demonstrated that after chemotherapy, various cancer cells release a significant number of EVs [8,9], which promote resistance to multiple drugs. For instance, the transfer of exosomes (a subgroup of EVs) from stromal cells to breast cancer cells enhances resistance to doxorubicin [10]. Additionally, the intercellular transfer of wild-type epidermal growth factor receptor (EGFR) via EVs induces resistance to osimertinib in non-small cell lung cancer (NSCLC) [11].

EVs contribute to cancer drug resistance by promoting drug efflux, apoptosis resistance, autophagy imbalance, and other prosurvival factors in cancer cells [4,5]. They also facilitate information exchange within the tumor microenvironment (TME), enabling signal transduction to fibroblasts, endothelial cells, immune cells, and other cells in the TME, thereby promoting phenotypic changes that support the development of cancer drug resistance [5]. Beyond the established understanding of how EVs promote resistance in target cells, new evidence suggests a link between EVs biogenesis and the development of drug resistance in donor cells. Furthermore, EVs can modulate the resistance of target cells by transporting and regulating noncoding RNAs (ncRNAs) to combat drug resistance. The contribution of EVs to cancer drug resistance can be categorized into two main aspects: (1) EVs biogenesis promotes the development of drug resistance in donor cells, and (2) EVs transfer facilitates the spread or inhibition of drug resistance in target cells. This dual role highlights the regulatory impact of EVs on target cell drug resistance. EVs play key roles in regulating cancer drug resistance.

This review explores the regulatory mechanisms of EVs biogenesis, secretion status, and cargo composition in relation to cancer drug resistance, providing a detailed examination of how these processes contribute to the development and maintenance of drug-resistant phenotypes in cancer cells. By examining recent findings and advancements, the review elucidates the complex interactions between EVs and the tumor microenvironment, highlighting potential therapeutic targets and strategies to mitigate EVs-mediated drug resistance. Proposed strategies include identifying drug resistance-inhibiting molecules within EVs, inhibiting EVs biogenesis and secretion, using EVs for drug delivery, and employing EVs to deliver resistance-inhibiting molecules. Aligned with the theme of extracellular vesicles as mediators and potential targets in combating cancer drug resistance, this review offers valuable insights for developing innovative strategies to effectively counteract cancer drug resistance.

## 2. Mechanisms by Which EVs Regulate Drug Resistance in Donor Cells

Upon exposure to cancer drugs, cancer cells release EVs, which foster drug resistance by increasing in count. After 24 h of exposure to doxorubicin, EVs in the plasma of tumor-bearing mice were significantly increased compared to controls [8]. Moreover, temozolomide-treated glioblastoma cells exhibited a significant increase in EVs count [12,13]. Docetaxel induced triple-negative breast cancer cells to secrete a greater number of sEVs [14]. In vitro experiments with NSCLC and chronic myeloid leukemia cells demonstrated that the number of EVs released by drug-resistant cancer cells was significantly greater than those released by drug-sensitive counterparts [15]. However, clinical studies have reported that androgen therapy affects the count and cargo of sEVs in prostate cancer patients [16], but there is currently no definitive evidence linking clinical androgen resistance to blood sEVs. The change in EVs count may be a fundamental characteristic of resistance [15,17,18]. To increase the reliability of the conclusions drawn between studies, it is necessary to reduce the heterogeneity of the techniques used. In addition, future studies should characterize distinct EVs subgroups released by cancer cells under various cancer drug treatments and consider utilizing the EVs count as a monitoring method for cancer drug therapy responses [5,19].

EVs as containers for drug efflux may be the mechanism by which the increased EVs count mediates the development of drug resistance in donor cells. The efflux of drugs from cells mediated by EVs represents one of the crucial mechanisms through which EVs regulate drug resistance in donor cells. When drugs are efficiently expelled from cells by EVs, the cells can survive. Studies have revealed that for certain cancer drugs, such as paclitaxel and cisplatin, 10–20% of the total drug dose can be loaded into EVs when donor cells are exposed to cancer drugs [20,21]. These studies, which involved breast and ovarian cancer cells under various conditions, demonstrated that cancer cells proficiently loaded paclitaxel into exosomes, with the amount reaching up to 10% of the cellular lysate under the same conditions [22]. The amount of paclitaxel in exosomes increased with increasing drug dose and prolonged exposure. Short-term exposure to docetaxel resulted in approximately 10% of the drug being detected in EVs derived from prostate cancer cells [23]. EVs derived from acute myeloid leukemia cells loaded with pegylated liposomal doxorubicin facilitated drug efflux and diminished the anticancer efficacy of the drug [24]. The release of tumor cell-derived EVs, which is facilitated by an acidic pH, accelerated the efflux of drugs such as cisplatin from cancer cells. Conversely, inhibiting EVs release using calpain inhibitors or specific small interfering RNAs (siRNAs) could triple the intracellular concentration of docetaxel in prostate cancer cells and reduce the necessary in vivo dose for anticancer effects [22]. Compared with their sensitive counterparts, drug-resistant cancer cells release EVs with a more potent payload of cancer drugs. Notably, EVs derived from breast cancer and acute lymphoblastic leukemia cells can accumulate anthracycline drugs, such as doxorubicin or pegylated liposomal doxorubicin, in a significantly greater number than those of their sensitive counterparts [22], which inhibits the release of extracellular vesicles, sensitizing cancer drugs [25]. These findings unequivocally demonstrated that EVs serve as effective mediators facilitating drug efflux, which represents a mechanism that contributes to the development of cancer drug resistance. Factors influencing the loading of cancer drugs into EVs include the drug exposure time, dosage, drug properties, and secretion speed. However, the specific mechanisms underlying drug loading into EVs have not been elucidated, and passive diffusion and the potential involvement of active transporters such as P-glycoprotein (P-gp) flipping on the EVs membrane or regulating the membrane lipid composition have been postulated [3]. An increase in the number of EVs may lead to an increase in drug loading within EVs. Moreover, EVs also have an efflux effect on targeted therapeutic drugs [26], and evidence for the drug efflux effect of EVs has been obtained. Both cell experiments and clinical trials have confirmed that higher drug loads in EVs are significantly correlated with poorer treatment responses [27]. Therefore, the drug loads in EVs may serve as a monitor for resistance in clinical practice. Given these insights, targeting EVs-mediated drug efflux could be a promising approach to combating drug resistance and improving cancer therapy outcomes.

## 3. Mechanisms by Which EVs Regulate Drug Resistance in Target Cells

### 3.1. Drug-Treated-Cells-Derived EVs Promote Survival or Drug Resistance in Target Cells

Upon exposure to cancer drugs, EVs derived from drug-treated cells intricately modulate target cell survival and the development of drug resistance. The application of EVs released by cisplatin-treated ovarian cancer cells to sensitive cells resulted in enhanced survival upon subsequent cisplatin treatment. Subsequent research indicated that the heightened survival capability of these cells might stem from the upregulation of the oncogenic gene p38 and the activation of the stress-responsive JNK signaling pathway induced by the transfer of EVs during cisplatin exposure [17]. The survival of bystander lung cancer cells exposed to cisplatin was enhanced by EVs secreted from cisplatin-treated lung cancer cells [28]. In vivo studies revealed the upregulation of miR-378a-3p and miR-378d in serum-derived EVs from breast cancer patients after docetaxel or paclitaxel treatment [29]. When introduced in vitro, these EVs cause cancer cells to become resistant to docetaxel or paclitaxel, and docetaxel-induced EVs promote docetaxel resistance in the tumor tissues of experimental mice [29]. At the molecular level, exposure to cancer drugs induces stress in cancer cells, resulting in increased apoptotic activity [30,31]. In some instances, apoptotic bodies (a subgroup of EVs, with sizes ranging from 50 to 5000 nm) can be detected, potentially representing an adaptation of tumor cells to drug toxicity and an adjustment of survival ability [31,32]. Consequently, EVs derived from drug-treated cancer cells play a role in promoting the survival and drug resistance of sensitive cells, underscoring the foundational role of EVs in regulating cell survival.

### 3.2. Resistant Cells-Derived EVs Promote Drug Resistance in Target Cells

EVs derived from drug-resistant cancer cells serve as mediators of drug resistance transmission to drug-sensitive cells. The drug resistance-promoting molecules contained in these EVs include ABC transporters, apoptosis resistance-related molecules, autophagy regulators, and various molecules intricately associated within the TME. When encapsulated in EVs, these molecules induce drug resistance in target cells through diverse mechanisms.

#### 3.2.1. Regulating Drug Transporters

ATP-binding cassette (ABC) transporters embedded in the membrane of EVs originating from cancer cells expedite the efflux of cancer drugs from target cells. ABC transporters, a conserved family involved in active transmembrane transport across various organisms, play a role in conferring resistance to cytotoxic and targeted drugs [33]. Proteomic analyses of EVs have revealed the presence of various ABC proteins, including ABCB1 (MDR1/P-gp), ABCC1 (MRP1), ABCC10 (MRP7), and ABCG2 (BCRP/MXR), all of which are widely implicated in drug resistance [33,34]. Notable examples have shown that EVs from prostate and lung cancer cells harboring ABCB1 mediate resistance to paclitaxel and docetaxel, respectively [35]. Breast cancer cell-derived EVs mediate paclitaxel resistance through ABCB2 [36]. ABCG2-encoded EVs derived from breast cancer cells facilitated the transmission of paclitaxel, doxorubicin, and topotecan [37]. Among the cargo of circular ncRNAs (circRNAs) in EVs, circ_0076305 targets and inhibits miR-186-5p, thereby augmenting the expression of the ABCC1 transporter protein and promoting cisplatin resistance in NSCLC cells [38]. Studies have confirmed that EVs can also transport ABCB1 transcriptional mRNA, enhancing the expression of ABCB1 in target cells [39]. However, the substrate specificity of ABC transporters was not highly selective [33], and there was no clear independent association with drug resistance in the clinical.

#### 3.2.2. Promoting Apoptosis Resistance

EVs from cancer cells encapsulate cargo that promotes apoptosis resistance. Apoptosis, catalyzed by the enzymatic activity of effector caspases, represents a form of cell death triggered by factors such as irreparable DNA damage, oxidative stress, cancer drugs, or adverse microenvironmental conditions. This process involves the modulation of DNA synthesis and repair, the use of cytotoxic drugs such as microtubule inhibitors, and various targeted therapies [40]. Apoptosis resistance is a significant mechanism of cancer drug resistance, and several cargos within EVs exacerbate this resistance. The apoptosis-inhibiting protein Survivin, a member of the inhibitor of apoptosis protein family, facilitates the transmission of paclitaxel and nocodazole resistance in breast cancer through exosomes [41]. The cancer proteins GSTP1 and STAT3 inhibited caspase cascades through EVs transmission and promoted resistance to 5-fluorouracil (5-FU) [42]. Transforming growth factor-β (TGF-β1)-mediated intercellular transfer through EVs leads to increased Smad2 phosphorylation and enhanced cell survival during drug treatment by inhibiting apoptosis and promoting cell motility [43]. The receptor tyrosine kinase TIE-1 reduces platinum sensitivity by inhibiting DNA-damage responses in recipient cells [44]. Plasma gelsolin, a plasma clotting protein, promoted resistance in ovarian cancer cells following exosomes transport. This effect was triggered by the upregulation of plasma GSN, which can promote the expression of the transcription factor HIF1α and suppress cisplatin-induced apoptosis. Increased levels of plasma gelsolin in ovarian cancer tissues are associated with a poorer clinical prognosis [45]. Cancer cells experience an increase in hypoxia-induced aerobic glycolysis and associated kinase pyruvate kinase isozyme type 2 (PKM2), which can potentially counteract platinum-induced reactive oxygen species (ROS) and indirectly inhibit apoptosis to promote cisplatin resistance [46]. PKM2 also reprograms cancer-associated fibroblasts (CAFs) and promotes an acidic TME, which enhances proliferation and cisplatin resistance in NSCLC cells [46]. Inhibiting the spread of drug resistance by suppressing apoptosis resistance promoted by EVs represents a promising approach to overcoming drug resistance in the clinical environment. Moreover, numerous molecules, such as the apoptosis inhibitor protein family and the p53 gene, which may potentially reverse cancer drug resistance, are worth exploring in the context of EVs promoting apoptosis resistance-mediated drug resistance [40].

#### 3.2.3. Regulating Autophagy

Cancer cells can exhibit imbalanced autophagy when exposed to drugs [47,48]. Autophagy, a process mediated by lysosomal degradation pathways, represents an adaptive cellular response. Alterations in signaling pathways and the regulation of the energy metabolism contribute to changes in autophagy levels, thereby facilitating the development of resistance to cancer drugs [49]. Some cargo within EVs can induce autophagy, consequently promoting drug resistance. For example, exosomes derived from hepatocellular carcinoma cells transport lysosome-associated membrane protein 2a (Lamp2a), and upregulating this protein would facilitate chaperone-mediated autophagy and enhance drug resistance [50]. An acidic microenvironment induces an increase in exosomal miR-21 levels and fosters malignant progression [51]. MiR-21, which inhibits autophagy through the PTEN/AKT pathway, is a critical factor in the acquisition of sorafenib resistance in liver cancer cells [51,52]. The serum-derived exosomal miRNAs miR-21-5p, miR-1246, miR-1229-5p, miR-135b, miR-425, and miR-96-5p serve as predictive biomarkers for drug resistance in advanced colorectal cancer patients. Bioinformatics analysis has suggested the involvement of these miRNAs in autophagy [53]. Similarly, circulating exosomal miR-425-3p, miR-1273h, miR-4755-5p, miR-9-5p, miR-146a-5p, and miR-215-5p are associated with autophagy and could act as biomarkers for cisplatin resistance in NSCLC [54]. Autophagy in cells and the release of EVs are likely to be synergistic with resistance to cancer drugs [47,48]. However, definitive findings regarding the association between autophagy and drug resistance are still limited, underscoring the need for further exploration of the relationship between autophagy in cancer cells and drug resistance (Figure 1).

#### 3.2.4. Mediating TME Communication

TME is considered a specialized niche that supports tumor initiation and progression, comprising the extracellular matrix, cancer cells, various normal cells, and an array of molecules such as cytokines, chemokines, and growth factors [55]. Information communication mediated by EVs not only regulates the TME but also influences the formation and dissemination of cancer drug resistance [56]. EVs impact metabolic reprogramming, phenotypic transition, and the transfer of procancer factors from TME constituent cells, concurrently modulating the physicochemical conditions of the TME to facilitate the development and spread of drug resistance. EVs play several crucial roles in TME information communication.

First, EVs can mediate nutrient metabolism reprogramming. EV-mediated information communication promotes nutrient metabolism reprogramming in TME constituent cells, suppressing aerobic respiration and enhancing anaerobic respiration. This reprogramming is a hallmark of cancer initiation and progression [55]. For instance, EVs derived from hepatocellular carcinoma cells transport the heat shock protein Hsp70, which increases reactive oxygen species levels and causes mitochondrial damage in recipient cells, thereby inhibiting aerobic respiration and promoting glycolysis [57]. Exosomes from hepatic stellate cells regulate colorectal tumor cell anaerobic respiration through the IL-6/STAT3 pathway, promoting resistance to irinotecan [58]. Aerobic glycolysis in cancer cells also leads to a decrease in the pH of endosomes and lysosomes, which facilitates EV acidification. Increased secretion of acidic EVs contributes to the resistance to hydrophobic cancer drugs [59].

Second, EVs facilitate the phenotypic transition of TME constituent cells. This transformation includes the activation and epithelialization of CAFs and the M2 polarization of macrophages, causing these cells to adopt more malignant phenotypes [55]. EVs derived from cancer cells mediate TGF-β-induced CAFs activation, while CAFs-derived exosomes promote anaerobic glycolysis, epithelial–mesenchymal transition (EMT), and resistance to anticancer drugs [60,61]. Various cytokines induce M2 polarization of macrophages, leading to immune suppression [62]. Exosomes derived from endothelial cells promote drug resistance in nasopharyngeal carcinoma cells, EMT in CAFs, and M2 polarization in tumor-associated macrophages [63]. Phenotypic transformation of TME constituent cells generally results in the development of cancer resistance and other conditions such as angiogenesis and metastasis [64].

Third, EVs can transfer procancer factors. The horizontal transfer of procancer factors within the TME significantly contributes to drug resistance. For example, the EV-mediated transport of EGFR promotes resistance to osimertinib in colorectal cancer [65]. PDGFRβ in EVs derived from BRAF inhibitor-resistant cells mediates the spread of resistance in melanoma, while isocitrate dehydrogenase 1 (IDH1) promotes resistance to 5-FU, and glutathione S-transferase P1 mediates resistance to doxorubicin [66,67]. PDGFRβ activates the PI3K/AKT pathway in a dose-dependent manner, facilitating BRAF inhibition [43,66]. The interplay between EVs and the TME in drug resistance is likely synergistic, as TME characteristics such as hypoxia and low pH promote the release of additional EVs. The roles of EVs and the TME in drug resistance are likely cooperative. These interactions between EVs and the TME emphasize their critical role in cancer progression and resistance mechanisms. Understanding these roles may lead to more effective cancer therapies. The regulatory effects and identified cargos of EVs for cancer drug resistance are summarized in Table 1, Table 2, Table 3 and Table 4.

### 3.3. Antiresistance Cargos That EVs Transported

Certain EVs originating from cancer cells or specific normal cells can suppress drug resistance in target cells. Currently, the identified mediators of drug resistance inhibition in such EVs primarily consist of miRNAs and circRNAs. The mechanisms by which these EVs exert inhibitory effects on drug resistance are as follows. The first category of EVs that can combat cancer drug resistance derives from normal cells or drug-sensitive cancer cells. These EVs-transported molecules suppress the survival of cancer cells. For example, miR-107, which regulates the mTOR pathway in gastric cancer cells, ultimately inhibits cancer cell proliferation. High miR-107 levels in gastric cancer cells through EVs could effectively reverse resistance to 5-fluorouracil and cisplatin [122]. The application of EVs derived from drug-sensitive cancer cells, which upregulate miR-613 in non-small cell lung cancer-resistant cells, could reverse resistance to cisplatin [134]. Similarly, the upregulation of miR-7 has a reversing effect on resistance to gefitinib [135]. In addition to miRNA, the circRNA FBXW7, which targets miR-128-3p, is highly expressed in normal colon cells and plays a role in reversing resistance to oxaliplatin when it is transported through EVs in colorectal cancer [169]. Furthermore, in comparison to that in drug-sensitive cancer cells, the downregulation of resistance-promoting molecules in EVs in resistant cancer cells, which could provide “negative signals”, may play inhibitory roles in drug resistance. Specific examples include miR-651 in cervical cancer [109], miR-770 in triple-negative breast cancer [107], miR-1915-3p in colorectal cancer [113].

Another source of EVs that inhibits target cell drug resistance is mesenchymal stem cells (MSCs). MSCs possess immune regulatory properties and exhibit homing and niche-like regenerative characteristics, suggesting that they are therapeutic agents for cancer [193,194]. MSCs have various sources, with bone marrow and the umbilical cord being common. EVs derived from bone marrow-derived MSCs could inhibit cisplatin resistance in non-small cell lung cancer and temozolomide resistance in glioblastoma, with key resistance-regulating factors identified being miR-193a and miR-199a, respectively [151,152]. On the other hand, EVs derived from umbilical cord MSCs could reverse imatinib resistance in chronic myeloid leukemia, with the identified resistance-regulating mediator being miR-145a-5p [153]. However, contrasting evidence suggests that EVs derived from bone marrow MSCs promote resistance to doxorubicin in colon cancer cells and contribute to Ara-C resistance in acute myeloid leukemia [150,195]. Therefore, the resistance-regulating effects of MSC-derived EVs vary across different tumors. While there may be promising applications for MSC-derived EVs, caution should be exercised when using these approaches [194].

Enhancing the content of drug resistance-inhibiting cargo in EVs may be an effective strategy for reversing drug resistance. Considering the widespread role of ncRNAs, the potential for discovering additional drug resistance-inhibiting ncRNAs in the future is obvious. The highlighted drug resistance-inhibiting cargos of EVs emphasize that the roles of EVs in combating drug resistance are not merely propagative but also inhibitory, underscoring the crucial regulatory roles of EVs in drug resistance (Figure 2). These insights pave the way for innovative therapeutic strategies that leverage the inhibitory potential of specific EVs to counteract drug resistance. Future research should focus on optimizing the delivery and efficacy of these EVs to maximize their therapeutic benefits in clinical settings.

## 4. Strategies for Combating Drug Resistance Through EVs

EVs have emerged as pivotal regulators of drug resistance, shedding light on innovative strategies to overcome drug resistance. This involves inhibiting EVs biogenesis and harnessing EVs for the delivery of cancer drugs or anti-resistance molecules.

### 4.1. Inhibiting EVs Biogenesis and Secretion

The use of inhibitors to curtail EVs biogenesis and secretion in cancer cells has proven effective in suppressing drug resistance. Multiple studies have substantiated the role of EVs in fostering drug resistance by inhibiting EVs biogenesis. The biogenesis of EVs is regulated by various factors such as the sorting of EVs cargo, cell type and condition, as well as the signals and stimuli for cells (Figure 1) [6]. The secretion of EVs is defined as an induced process in which cells participate in response to stimuli or pathological conditions [6]. Both the non-specific consumption of cholesterol and the inhibition of microenvironment acidification are methods to suppress EVs secretion [196,197]. In addition, some carcinogenic factors also influence the biogenesis and secretion of EVs. For instance, the overexpression of polymerase I and transcript release factor (PTRF) are a pro-carcinogenic factor, and its upregulation in cancer cells is significantly correlated with enhanced EVs secretion. Knocking down PTRF not only reduces EVs secretion but also inhibits the drug resistance of glioblastoma exposed to temozolomide [198]. When leukemia cells are exposed to tyrosine kinase inhibitors, the pro-carcinogenic factor fibroblast growth factor two is associated with an increase in exosomes and plays a role in promoting drug resistance [199]. Therefore, targeting the biogenesis process of EVs or pro-carcinogenic factors associated with EVs secretion can reduce EVs and inhibit drug resistance [196]. This inhibition not only reduced EVs-mediated drug efflux but also disrupted intercellular communication facilitated by EVs. Identification of potent inhibitors for EVs biogenesis is a critical research objective. Drugs such as GW4869 [192,196], GW4876, ketoconazole, calpain inhibitors, and chloroquine [111] have demonstrated noteworthy inhibitory effects on EVs release. Importantly, these molecules impeded EVs biogenesis and dampened drug resistance [200,201]. Identifying highly effective inhibitors with minimal toxicity to normal cells is urgently needed.

### 4.2. Engineered EVs as Transporters

Engineered EVs can serve as effective drug delivery transporters. Engineered EVs are typically derived from carefully selected natural EVs that have been remolded [202]. They have prolonged circulating times in vivo and could be absorbed and utilized by cells [203]. Engineered EVs can effectively overcome drug resistance by loading cancer drugs and specific resistance antagonists [202,203]. According to the mechanisms that EVs use to regulate resistance, the selective loading of cargo is a key issue in engineering EVs to combat drug resistance.

#### 4.2.1. Transporting Drugs

The utilization of EVs for loading anticancer drugs holds promise for improving the bioavailability of cancer drugs. In the context of research on the mechanisms associated with drug resistance, it has been observed that EVs loaded with anticancer drugs retain their anticancer efficacy upon transport. For instance, when EVs are loaded with paclitaxel, secreted by cancer cells after exposure to paclitaxel and subsequently applied to adjacent cancer cells, these vesicles exhibit no proliferative effects. The anticancer effect of paclitaxel within vesicles was more potent than that of its original form in some cases [22,204]. Independent studies have shown that EVs can enhance the solubility of paclitaxel in water. Therefore, delivering cancer drugs such as paclitaxel through EVs may constitute a strategy to enhance the in vivo utilization efficiency of paclitaxel. Once anticancer drugs are loaded into EVs, they can be efficiently absorbed by cells. However, the intracellular drug concentrations in the target cells were significantly elevated, demonstrating the effective delivery of anticancer drugs by EVs. In vitro experiments have revealed that extracellular vesicles derived from prostate cancer cells or melanoma cells serve as excellent carriers of paclitaxel. The enrichment effect of these vesicles resulted in a significantly increase in the solubility of paclitaxel in water [59,205]. The use of milk-derived extracellular vesicles achieved drug loading capacities of approximately 30% for paclitaxel and 20% for 5-FU [206]. Compared to the simple diffusion and passive transport of anticancer drugs across cell membranes in vivo, EVs carrying anticancer drugs might benefit from a membrane fusion mechanism and enhanced drug absorption. Extracellular vesicles derived from bone marrow mesenchymal stem cells achieved encapsulation efficiencies of approximately 8% for gemcitabine and approximately 2% for paclitaxel [207]. Administering these EVs-loaded drugs into experimental mice demonstrated a more effective anticancer outcome than administering the naked anticancer drugs. The use of EVs for the repackaging of anticancer drugs facilitates drug modification and targeted absorption [208]. Studies conducted in zebrafish have shown that paclitaxel or doxorubicin can traverse the blood–brain barrier via the transport of extracellular vesicles, effectively achieving an anticancer effect against brain tumors [209]. In summary, EVs present a highly promising approach for improving drug bioavailability and targeting. It is crucial to select nontumorigenic EVs as in vivo drug delivery carriers to avoid potential risks. Sources such as EVs derived from normal cells or plant-derived EVs [210], like bitter melon and skimmed milk, may offer benefits for anticancer treatments considering the various anticancer properties associated with plant-derived methods [211,212].

#### 4.2.2. Transporting ncRNAs

Artificially loading resistance-inhibiting ncRNAs into EVs is a potent strategy for combating cancer drug resistance. The most significant mechanism through which miRNAs in EVs inhibit drug resistance involves promoting apoptosis or inhibiting autophagy. For instance, miR-204-5p might suppress exosomes biogenesis, promote apoptosis, and inhibit resistance to 5-FU in gastric cancer cells [213]. Similarly, the loading of miR-134 in EVs enhances apoptosis and suppresses cisplatin resistance in triple-negative breast cancer cells [105]. The inhibition of autophagy is another crucial molecular mechanism through which miRNAs from specific EVs exert their inhibitory effect on drug resistance. In addition to well-established molecules known for suppressing the resistance of EVs, EVs can also be loaded with molecules that inhibit drug resistance-promoting molecules. A study demonstrated that introducing a miR-15 inhibitor through EVs into oral squamous cell carcinoma cells could effectively reverse resistance to cisplatin [108]. Loading EVs with a miR-214 inhibitor could also reverse gastric cancer cell resistance to cisplatin [142]. Inhibitory siRNAs targeting well-validated oncogenes could also be delivered through EVs to resist drug resistance. Using EVs to transport ncRNAs to reverse drug resistance is a noteworthy consideration in reversing drug resistance.

#### 4.2.3. Cotransporting Drugs and Resistance Inhibitors

Simultaneously, loading cancer drugs and resistance-inhibiting molecules into EVs may be an effective strategy for preventing cancer drug resistance. The combination of anticancer drugs and resistance-inhibiting molecules in EVs is currently a widely validated codelivery strategy. For example, carnitine palmitoyltransferase 1A (CPT1A) is a key molecule that promotes oxaliplatin resistance in colorectal cancer. Loading oxaliplatin and a CPT1A inhibitor such as etomoxir into EVs could inhibit resistance, which would enable oxaliplatin to have a powerful cytotoxic effect on tumor cells [214]. Similar examples include combinations such as doxorubicin and MRP1-siRNA [215], cabazitaxel and TNF-related apoptosis-inducing ligand (TRAIL) [216], oxaliplatin and PGM5-AS1 [217], temozolomide and angiotensinogen (AGT) [218], and paclitaxel or cisplatin combined with Survivin inhibitors [219]. These combinations significantly inhibited drug resistance. The simultaneous delivery of anticancer drugs and resistance-inhibiting small molecules through EVs has broad feasibility and promising efficacy. Future research may include extensive experiments to determine the effectiveness of drug combinations involving anticancer drugs and corresponding resistance-inhibitory molecules delivered through EVs (Figure 3). The interaction between EVs and cells can either involve fusion with the cell membrane, releasing their contents into the cytoplasm, or internalization into endosomes and/or other intracellular compartments without fusion. These effects of EVs on recipient cells may be mediated by binding to receptors on the cell surface or internally, or by the release of their contents [6,7]. However, the use of EVs as drug delivery strategies may face challenges such as inefficient isolation, high heterogeneity, low yield, and biological variability, which limit large-scale production and a standardized application. Addressing these issues can further enhance the clinical application potential of EVs and reduce their limitations in drug delivery. Moreover, considering the regulating role of EVs in drug resistance, EVs as drug delivery strategies need to be carefully controlled.

## 5. Conclusions and Outlook

Undoubtedly, EVs are proving instrumental in unraveling the complex mechanisms behind drug resistance in cancer treatment. Their dual role—both in promoting and combating drug resistance—places EVs at the forefront of innovative therapeutic targets. Predominantly, EVs contribute to drug resistance by expelling cancer drugs from cells and mediating intercellular communication that supports tumor survival and adaptation. However, recent research highlights the potential of EVs to carry resistance-inhibiting molecules, offering a novel approach to counteract this resistance.

One of the most promising directions for overcoming drug resistance lies in the ncRNAs present in EVs. These ncRNAs have shown significant potential in regulating cancer cell survival, promoting apoptosis, and inhibiting autophagy, thus reversing resistance to various anticancer drugs. Despite these promising findings, extensive studies are necessary to determine the clinical feasibility of EVs-based strategies for combating drug resistance. This includes validating the therapeutic potential of ncRNA-loaded EVs and ensuring their safety and efficacy in clinical settings. Future research must address several technical limitations that currently hinder the consistent application of EVs-based therapies. Standardizing the preparation and isolation methods of EVs is crucial to ensure reproducibility and reliability in clinical practice. Furthermore, the use of EVs from non-tumorigenic sources, such as normal cells or plant-derived EVs, could enhance the safety and applicability of these treatments. In conclusion, EVs represent a groundbreaking avenue in cancer therapy improvement, with the potential to revolutionize how drug resistance is managed. By continuing to explore and refine the use of EVs, the medical community may develop more effective, personalized treatment options that improve patient outcomes and provide a reference for rational cancer therapy. The future of EVs-based therapies looks promising, with the potential to make significant strides in the battle against cancer drug resistance.

## Figures and Tables

**Figure 1 molecules-30-00498-f001:**
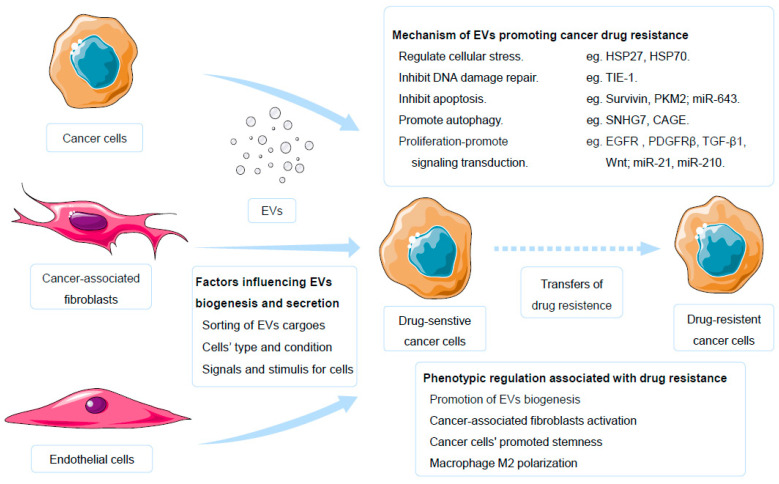
EVs promote cancer drug resistance. EVs, extracellular vesicles.

**Figure 2 molecules-30-00498-f002:**
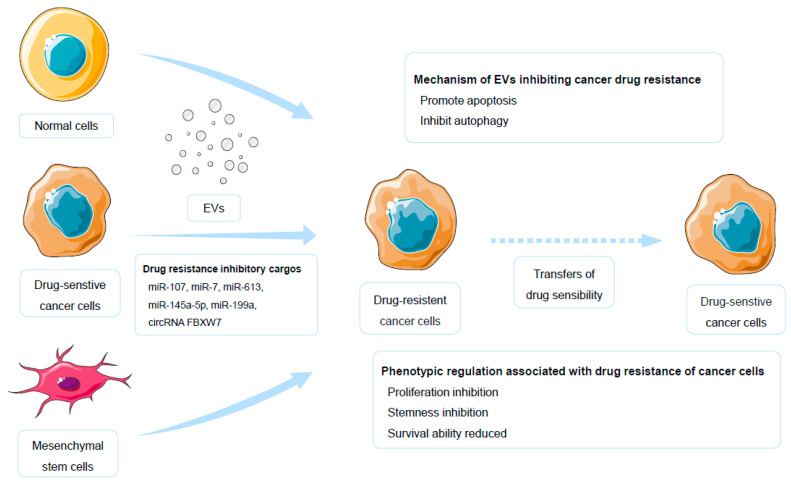
EVs inhibit cancer drug resistance. EVs, extracellular vesicles.

**Figure 3 molecules-30-00498-f003:**
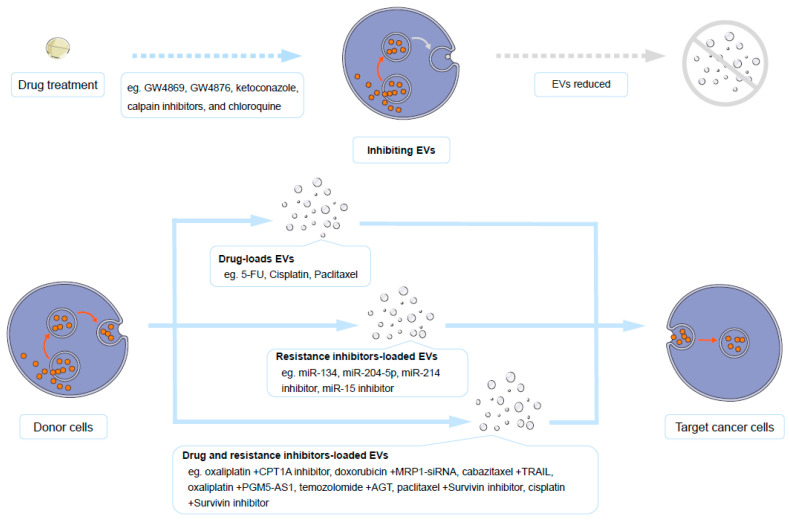
Strategies to combat cancer drug resistance through EVs. EVs, extracellular vesicles. 5-FU, 5-fluorouracil.

**Table 1 molecules-30-00498-t001:** Protein cargos of EVs.

Sources	Effects	Drugs	Cargos	Molecular Mechanisms	Regulations	References
Breast cancer #	Promote	--	TrpC5	--	Up	[68]
Breast cancer	Promote	Bexarotene + Guggulsterone, Doxorubicin	ABCG2	--	Up	[69]
Breast cancer #	Promote	CDK4/6 inhibitors	TK1, CDK9	--	Up	[70]
Breast cancer #	Promote	Doxorubicin	CD44	--	Up	[71]
Breast cancer #	Promote	Doxorubicin	GSTP1	--	Up	[42]
Breast cancer	Promote	Doxorubicin	HSP70	promotes reactive oxygen and mitochondrial damage in target cells, inhibits aerobic respiration, and promotes glycolysis	Up	[72]
Breast cancer	Promote	Doxorubicin	TGF-β1	regulates Smad2 which inhibits apoptosis and enhances cells’ mobility	Up	[43]
Breast cancer #	Promote	Doxorubicin	UCH-L1	activates the MAPK/ERK signaling pathway	Up	[73]
Breast cancer	Promote	Doxorubicin	TrpC5	influences EVs biogenesis and promotes target cells’ P-gp	Up	[74]
Breast cancer	Promote	Paclitaxel	Survivin	promotes apoptotic resistance	Up	[75]
Triple negative breast cancer	Promote	Gemcitabine	Annexin A6	inhibits the ubiquitination and degradation of EGFR	Up	[76]
Cervix cancer #	Promote	Cisplatin	RAB7A	influences EVs biogenesis	Down	[77]
Colon cancer	Promote	Gefitinib	CD133	regulates KRAS’s downstream signal transduction	Up	[78]
Colorectal cancer	Promote	5-Fluorouracil	GSTP1, p-STAT3	regulate the activities of the small guanosine 5′-triphosphatases RhoA and Rac1	Up	[79]
Colorectal cancer	Promote	5-Fluorouracil	IDH1	decreases the level of NADPH	Up	[67]
Diffuse large B-cell lymphoma #	Promote	Cyclophosphamide, Doxorubicin, Vincristine, Prednisone, Rituximab	CA1	regulates the NF-kB and STAT3 signaling pathways	Up	[80]
Gastric cancer	Promote	Cisplatin	RPS3	acts through PI3K-Akt-cofilin-1 signaling pathway	Up	[81]
Gastric cancer	Promote	Doxorubicin, Paclitaxel	CAGE	increases autophagy	Up	[82]
Gastric cancer	Promote	Vincristine	CLIC1	may relate to upregulate P-gp and Bcl-2	Up	[83]
Glioblastoma	Promote	Temozolomide	HSP27	--	Up	[84]
Glioma	Promote	Temozolomide	Connexin 43	--	Up	[85]
Glioma	Promote	Temozolomide	TIMP3	regulates immunity through the TIMP3/PI3K/AKT axis	Up	[86]
Liver cancer	Promote	5-Fluorouracil	RAB27B	influences EVs biogenesis	Up	[87]
Liver cancer #	Promote	5-Fluorouracil, Cabozantinib S-malate, Docetaxel, Doxorubicin, Irinotecan hydrochloride, Lenvatinib mesylate, Suberoylanilide hydroxamic acid, Sorafenib tosylate, Regorafenib	RAB3B	influences EVs biogenesis	Up	[88]
Liver cancer	Promote	Cisplatin	Lamp2a	influences EVs biogenesis and upregulates apoptosis	Up	[50,57]
Lung cancer	Promote	Docetaxel	SNHG7	induces autophagy and macrophage M2 polarization	Up	[89]
Non-small cell lung cancer	Promote	Cisplatin	PKM2	promotes intracellular glycolysis, may neutralize reactive oxygen, inhibit apoptosis, reprogram CAFs, and produce acidic TME	Up	[46]
Non-small cell lung cancer	Promote	Osimertinib	EGFR	upregulates Rab GTP enzyme	Up	[10]
Melanoma	Promote	BRAF inhibitors	PDGFRβ	regulates PI3K/AKT signaling pathway	Up	[43,66]
Melanoma #	Promote	Puromycin	PD-L1	regulates immunity reactions	Up	[90]
Ovarian cancer #	Promote	Cisplatin	pGSN	--	Up	[45]
Ovarian cancer	Promote	Cisplatin	O-GlcNAcylation of SNAP-23	influence EVs biogenesis	Down	[91]
Ovarian cancer	Promote	Cisplatin	TIE-1	inhibits DNA damage response	Up	[44]
Ovarian cancer #	Promote	Platinum compounds	CLPTM1L/CRR9	--	Up	[92]
Pancreatic cancer	Promote	Gemcitabine	EphA2	--	Up	[93]
Pancreatic ductal cancer	Promote	Gemcitabine	MMP14	mediates cells’ adhesion, promotes cells’ ability to form pellets, and migrating	Up	[94]
Prostate cancer #	Promote	Enzalutamide	Syntaxin 6	influences EVs biogenesis	Up	[95]
Renal cancer	Promote	Sunitinib	RAB27B	influences EVs biogenesis	Up	[96]
Tumor stem cells (Liver cancer)	Promote	Regorafenib	RAB27B	influences EVs biogenesis	Up	[97]
Endothelial cells (Umbilical cord), Colorectal cancer, Ovarian cancer #	Promote	Anti-VEGF	CD63	influences EVs biogenesis	Up	[26]
Cancer-associated fibroblasts	Promote	5-Fluorouracil	Wnt	influences EVs biogenesis, promotes Wnt signaling pathway, and promotes drug resistance in colorectal cancer	Up	[98]
M2 macrophages #	Promote	Cisplatin	GATA3	regulates transcription and promotes drug resistance in high-grade serous ovarian carcinoma	Up	[99]

EVs, extracellular vesicles. #, Results were supported by clinical evidence. Effects, EVs’ regulation effects on drug resistance. --, Details were not provided.

**Table 2 molecules-30-00498-t002:** MicroRNA cargos of EVs.

Sources	Effects	Drugs	Cargos	Molecular Mechanisms	Regulations	References
Breast cancer	Promote	Doxorubicin, Paclitaxel	miR-155	--	Up	[100]
Breast cancer	Promote	Tamoxifen	miR-205	targets E2F1	Up	[101]
Breast cancer	Promote	Tamoxifen	miR-9-5p	downregulates ADIPOQ	Up	[102]
Breast cancer #	Promote	CDK4/6 inhibitors	miR-432-5p	targets TGF-β pathway and promotes Cdk6	Up	[103]
Breast cancer #	Promote	Doxorubicin	miR-222	targets PTEN/AKT and induces macrophage M2 polarization	Up	[104]
Breast cancer #	Promote	Paclitaxel	miR-378a-3p, miR-378d	activate EZH2/STAT3 signaling pathway, promotes cancer cell stemness and drug resistance	Up	[29]
Breast cancer, Colorectal cancer, Gastric cancer, Glioma, Lung cancer (loaded drug resistance inhibitor)	Inhibit	5-Fluorouracil	miR-204-5p	inhibits RAB22A and Bcl2	Up	[105]
Triple negative breast cancer	Promote	Cisplatin	miR-423-5p	--	Up	[106]
Triple negative breast cancer #	Promote	Doxorubicin	miR-770	targets to STMN1, inhibits EMT and the migration and invasion of cancer cells	Down	[107]
Triple negative breast cancer (Loaded drug resistance inhibitor)	Inhibit	Cisplatin	miR-134	upregulates STAT5b, Hsp90, and Bcl-2, inhibits proliferation and drug resistance	Up	[108]
Cervical cancer #	Promote	Cisplatin	miR-651	directly targets ATG3	Down	[109]
Chronic myeloid leukemia	Promote	Imatinib	miR-365	--	Up	[110]
Colon cancer	Promote	5-Fluorouracil	miR-21	targets PDCD4, TPM1, and PTEN	Up	[51,52]
Colorectal cancer	Promote	Oxaliplatin	miR-19b	downregulates AGAP2	Up	[111]
Colorectal cancer	Promote	Oxaliplatin	miR-46146	targets PDCD10	Up	[112]
Colorectal cancer #	Promote	Oxaliplatin	miR-1915-3p	promotes PFKFB3, USP2, and E-cadherin; downregulates EMT	Down	[113]
Diffuse large B-cell lymphoma	Promote	Rituximab	miR-125b-5p	targets TNFAIP3	Up	[114]
Esophageal cancer	Promote	Cisplatin	miR-193	--	Up	[115]
Esophageal cancer	Promote	Cisplatin	miR-21	targets PDCD4	Up	[116]
Gastric cancer	Promote	5-Fluorouracil	miR-106a-5p, miR-421	influences hypermethylation of TFAP2E	Up	[117]
Gastric cancer	Promote	Cisplatin	miR-500a-3p	downregulates FBXW7, promotes stemness and drug resistance of cancer cells	Up	[118]
Gastric cancer	Promote	Doxorubicin	miR-501	downregulates BLID, leading to inactivation of caspase-9/-3 and phosphorylation of Akt	Up	[119]
Gastric cancer	Promote	Paclitaxel	miR-155-5p	targets GATA, TP53INP1	Up	[120]
Gastric cancer	Promote	Trastuzumab	miR-301a-3p	induces endoplasmic reticulum stress	Up	[121]
Gastric cancer (Drugs sensitive cells)	Inhibit	5-Fluorouracil, Cisplatin	miR-107	downregulates HMGA2 and inhibits MGA2/mTOR/P-gp pathway	Up	[122]
Gastric cancer #	Promote	5-Fluorouracil	miR-374a-5p	increases NeuroD1 and promotes apoptosis resistance	Up	[123]
Glioblastoma	Promote	Temozolomide	miR-25-3p	regulates Cyclin E	Up	[124]
Glioblastoma multiforme	Promote	Temozolomide	miR-93, miR-193	target Cyclin D1	Up	[125]
Glioblastoma #	Promote	Temozolomide	miR-1238	--	Up	[126]
Liver cancer	Promote	Cisplatin	miR-106a/b	targets SIRT1	Down	[127]
Liver cancer	Promote	Cisplatin	miR-643	regulates APOL6 and leads to apoptosis resistance	Up	[128]
Liver cancer #	Promote	Sorafenib	miR-744	targets PAX2	Down	[129]
Lung cancer	Promote	Gefitinib	miR-21	--	Up	[130]
Non-small cell lung cancer	Promote	Anlotinib	miR-136-5p	activates the Akt pathway by targeting PPP2R2A	Up	[131]
Non-small cell lung cancer	Promote	Gefitinib	miR-214	--	Up	[132]
Non-small cell lung cancer	Promote	Osimotinib	miR-210-3p	--	Up	[133]
Non-small cell lung cancer (Drug sensitive cells)	Inhibit	Cisplatin	miR-613	downregulates of GJA1, TBP, and EIF-4E	Up	[134]
Non-small cell lung cancer (Drug sensitive cells) #	Inhibit	Gefitinib	miR-7	targets YAP	Up	[135]
Non-small cell lung cancer #	Promote	Cisplatin	miR-1273a	regulates SDCBP	Down	[136]
Non-small cell lung cancer #	Promote	Cisplatin	miR-21	--	Up	[137]
Non-small cell lung cancer #	Promote	Cisplatin	miR-425-3p	targets AKT1 and promotes autophagy	Up	[138]
Non-small cell lung cancer #	Promote	Cisplatin	miR-4443	targets METTL3	Up	[139]
Small cell lung cancer #	Promote	Platinum compounds	miR-92b-3p	acts through the PTEN/AKT pathway	Up	[140]
Nasopharyngeal carcinoma	Promote	Cisplatin	miR-106a-5p	targets ARNT2 and activates AKT phosphorylation	Up	[141]
Oral squamous cell carcinoma (Loaded drug resistance inhibitor)	Inhibit	Cisplatin	miR-15 inhibitor	downregulates FOXO3A and promotes EMT	Up	[142]
Ovarian cancer	Promote	Cisplatin	miR-21-5p	inhibits PDHA1, which promotes cancer cells survival and glycolysis	Up	[143]
Ovarian cancer #	Promote	Cisplatin	miR-429	acts through CASR/STAT3 pathway	Up	[144]
Prostate cancer	Promote	Cisplatin, Doxorubicin, Docetaxel	miR-27a	inhibits P53	Up	[145]
Renal cancer #	Promote	Sorafenib	miR-31-5p	directly targets MLH1	Up	[146]
Tongue squamous cell carcinoma	Promote	Docetaxel	miR-200c	regulates TUBB3 and PPP2R1B	Down	[147]
Tumor stem cells (Pancreatic cancer)	Promote	Gemcitabine	miR-210	may activate mTOR signaling pathway	Up	[148]
Endothelial cells (Dermis microvascular)	Promote	5-Fluorouracil	miR-1246	induces IL-6	Up	[149]
Mesenchymal stem cells (Bone marrow)	Promote	Doxorubicin	miR-142-3p	regulates Notch pathway, enhances stemness and drug resistance in colon cancer cells	Up	[150]
Mesenchymal stem cells (Bone marrow) #	Inhibit	Cisplatin	miR-193a	targets LRRC1 and inhibits drug resistance in non-small cell lung cancer cells	Down	[151]
Mesenchymal stem cells (Bone marrow) #	Inhibit	Temozolomide	miR-199a	downregulates AGAP2 and inhibits drug resistance in glioma	Up	[152]
Mesenchymal stem cells (Umbilical cord) #	Inhibit	Imatinib	miR-145a-5p	targets GLS1 to increase GLS protein, which inhibits GLS1 ubiquitination to promote IM-induced apoptosis in chronic myeloid leukemia	Up	[153]

EVs, extracellular vesicles. #, Results were supported by clinical evidence. Effects, EVs’ regulation effects on drug resistance. --, Details were not provided.

**Table 3 molecules-30-00498-t003:** CircRNA cargos of EVs.

Sources	Effects	Drugs	Cargos	Molecular Mechanisms	Regulations	References
Breast cancer #	Promote	Lapatinib	circ-MMP11	regulates miR-153-3p/ANLN axis	Up	[154]
Breast cancer #	Promote	Tamoxifen	circ_UBE2D2	interacts with miR-200a-3p	Up	[155]
Colorectal cancer #	Promote	5-Fluorouracil	circ_0000338	regulates miR-217 and miR-485-3p	Up	[156]
Colorectal cancer #	Promote	Oxaliplatin	circ_0005963 (circ-122)	acts as a sponge for miR-122 to regulate PKM2	Up	[157]
Esophageal cancer #	Promote	Cisplatin	circ_0000337	acts as a sponge for miR-377-3p to regulate JAK2	Up	[158]
Gastric cancer	Promote	Cisplatin	circ-PVT1	targets miR-30a-5p and affects apoptosis, invasion, or autophagy	Up	[159]
Gastric cancer	Promote	Oxaliplatin	circ_0032821	acts as a sponge for miR-515-5p to regulate SOX9	Up	[160]
Gastric cancer #	Promote	Cisplatin	circ_0000260	targets miR-129-5p to influence the development and deterioration of gastric adenocarcinoma by upregulating MMP11	Up	[161]
Gastric cancer #	Promote	Doxorubicin	circ_PRRX1	targets miR-3064-5p to regulate PTPN14	Up	[162]
Glioma	Promote	Temozolomide	circ-GLIS3	upregulates MED31 by sponging miR-548m	Up	[163]
Glioma #	Promote	Temozolomide	circ_0072083	regulates expressions of ALKBH5 and ANOG through miR-1252-5p	Up	[164]
Liver cancer #	Promote	Anti-PD1	circUHRF1 (circ _0048677)	regulates miR449c-5p which upregulates TIM-3	Up	[165]
Neuroblastoma	Promote	Doxorubicin	circ_DLGAP4	regulates miR-143 to HK2, enhances glycolysis and drug resistance	Up	[166]
Non-small cell lung cancer	Promote	Cisplatin	circ_0076305	induces miR-186-5p to enhance ABCC1	Up	[38]
Non-small cell lung cancer #	Promote	Cisplatin	circ_0014235	regulates miR-520a-5p/CDK4 axis	Up	[167]
Non-small cell lung cancer #	Promote	Cisplatin	circ_PIP5K1A	acts through miR-101/ABCC1 axis	Up	[168]
Non-small cell lung cancer #	Promote	Cisplatin	circ-CPA4	acts as a let-7 miRNA sponge to downregulate PD-L1	Up	[169]
Non-small cell lung cancer #	Promote	EGFR-TKIs	circRNA_102481	acts as a sponge for miR-30a-5p which regulates ROR1	Up	[170]
Oral squamous cell carcinoma #	Promote	Cisplatin	circ-SCMH1 (circ_0011946)	acts as a sponge of miR-338-3p which regulates LIN28B	Up	[171]
Osteosarcoma #	Promote	Cisplatin	circ_103801	--	Up	[172]
Ovarian cancer #	Promote	Cisplatin	circ_C20orf11	regulates miR-527/YWHAZ axis	Up	[173]
Prostate cancer #	Promote	Docetaxel	circ_XIAP	acts through miR-1182/TPD52 axis	Up	[174]
Normal colon #	Inhibit	Oxaliplatin	circRNA FBXW7	directly binds to miR-128-3p and inhibits apoptosis and EMT	Up	[169]

EVs, extracellular vesicles. #, Results were supported by clinical evidence. Effects, EVs’ regulation effects on drug resistance. --, Details were not provided.

**Table 4 molecules-30-00498-t004:** LncRNA cargos of EVs.

Sources	Effects	Drugs	Cargos	Molecular Mechanisms	Regulations	References
Breast cancer	Promote	Tamoxifen	lncRNA UCA1	--	Up	[175]
Breast cancer	Promote	Trastuzumab	AGAP2-AS1	--	Up	[176]
Breast cancer #	Promote	Doxorubicin	lncRNA H19	--	Up	[177]
Breast cancer #	Promote	Trastuzumab	OIP5-AS1	targets miR-381-3p which regulates HMGB3	Up	[178]
Esophageal cancer	Promote	Doxorubicin	linc-VLDLR	--	Up	[179]
Esophageal squamous cell carcinoma #	Promote	Cisplatin	lncRNA POU3F3	--	Up	[180]
Gastric cancer #	Promote	Cisplatin	lncRNA HOTTIP	acts through the miR-218/HMGA1 axis	Up	[181]
Glioblastoma	Promote	Temozolomide	lncRNA HOTAIR	acts through the miR-519a-3p/RRM1 axis	Up	[182]
Glioblastoma #	Promote	Temozolomide	lncRNA SBF2-AS1	acts as a competing endogenous RNA of miR-151a-3p, which leads to the disinhibition of XRCC4 and enhances DNA double-strand break repair in tumor cells	Up	[183]
Lung cancer #	Promote	--	lncRNA PCAT1	targets miR-182/miR-217 which regulates p27/CDK6	Up	[184]
Non-small cell lung cancer	Promote	EGFR-TKIs	lncRNA SOX2	targets miR-627-3p upregulated Smads, promoting drug resistance and macrophage M2 polarization	Up	[185]
Non-small cell lung cancer	Promote	Erlotinib	lncRNA H19	acts through miR-615-3p/ATG7 axis	Up	[186]
Oral squamous cell carcinoma #	Promote	5-Fluorouracil	APCDD1L-AS1	targets miR-1224-5p which regulates NSD2	Up	[187]
Osteosarcoma #	Promote	Doxycycline	lncRNA ANCR	--	Up	[188]
Ovarian cancer #	Promote	Cisplatin	lncRNA UCA1	acts through miR-143/FOSL2 axis	Up	[189]
Squamous cell carcinoma of skin #	Promote	Cisplatin	lnc-PICSAR	acts through the miR-485-5p/REV3L axis	Up	[190]
Tongue squamous cell carcinoma	Promote	Cisplatin	lncRNA HEIH	acts as a ceRNA of miR-3619-5p, upregulates HDGF which promotes proliferation, drug resistance, and inhibits cell apoptosis	Up	[191]
Cancer-associated fibroblasts #	Promote	Oxaliplatin	lncRNA CCAL	activates β-catenin pathway	Up	[192]

EVs, extracellular vesicles. #, Results were supported by clinical evidence. Effects, EVs’ regulation effects on drug resistance. --, Details were not provided.

## Data Availability

There is no data associated with the manuscript.

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
