# Peer review of "Extracellular Vesicles as Mediators and Potential Targets in Combating Cancer Drug Resistance"

_molecules, 2025, doi:10.3390/molecules30030498_

Round 1
Reviewer 1 Report
Comments and Suggestions for Authors
In the present review, Zhang et al. demonstrate the potential role of EVs as mediators and targets against drug resistance. The review is well-written with significant interests to the readers in the field. However, a few minor modifications could improve the article further. My comments are as follows:
1. As the review discusses the role of EVs in drug resistance, a in brief description of EVs should be provided.
2. Please include the current MISEV classification of the EVs in brief.
3. The principal drawbacks on the role of EVs as drug delivery vehicles should also be discussed thoroughly.
4. Please include the possibility of targeting EV fusion mechanism with the recipient cells as the strategy for combating drug resistance through EVs.
Comments on the Quality of English Language
Minor editing is required.
Author Response
Reviewer 1
In the present review, Zhang et al. demonstrate the potential role of EVs as mediators and targets against drug resistance. The review is well-written with significant interests to the readers in the field. However, a few minor modifications could improve the article further. My comments are as follows:
- As the review discusses the role of EVs in drug resistance, a in brief description of EVs should be provided.
Response: Thank you sincerely for your reminder. The Chapter 1. Introduction of our manuscript offers a concise overview of EVs, and the content included is essential to our topic. Adding more details may be redundant. The details in the manuscript are: "Extracellular vesicles (EVs), as mediators in the communication between cancer cells and their microenvironment, dynamically regulate the survival and signaling pathways of cells through various mechanisms [3, 4], which leads to resistance to multiple cancer drugs [5]."
- Please include the current MISEV classification of the EVs in brief.
Response: Thank you very much for your insightful reminder. After carefully reading the literature of MISEV, we revised the content and added the MISEV classification of the EVs.
Orginal contents of our manuscript: "EVs are membrane vesicles secreted by cells that include exosomes and microvesicles that originate from late endosomes and the plasma membrane, with sizes ranging from 50-150 nm and 50-500 nm, respectively [7]."
The contents have been changed to: "EVs are generally classified into large EVs (diameter >200 nm) and small EVs (sEVs,diameter <200 nm). Exosomes, a subset of sEVs, originate from the endosomal system and are released through the multivesicular body (MVB) pathway, whereas microvesicles, derived from the plasma membrane, typically refer to larger EVs or all EVs [6, 7]."
We have also added references of MISEV: "7.Welsh JA, Goberdhan DCI, O'Driscoll L, et al. Minimal information for studies of extracellular vesicles (MISEV2023): From basic to advanced approaches [published correction appears in J Extracell Vesicles. 2024 May;13(5):e12451. doi: 10.1002/jev2.12451]. J Extracell Vesicles. 2024;13(2):e12404. doi:10.1002/jev2.12404".
- The principal drawbacks on the role of EVs as drug delivery vehicles should also be discussed thoroughly.
- Please include the possibility of targeting EV fusion mechanism with the recipient cells as the strategy for combating drug resistance through EVs.
Response: Thank you very much for your kind reminder. We have added description of these two issues in the manuscript.
Added the following content to section 4.2.3. Cotransporting drugs and resistance inhibitors:
"The interaction between EVs and cells can either involve fusion with the cell membrane, releasing their contents into the cytoplasm, or internalization into endosomes and/or other intracellular compartments without fusion. These effects of EVs on recipient cells may be mediated by binding to receptors on the cell surface or internally, or by the release of their contents [6, 7]. However, the use of EVs as drug delivery strategies may face challenges such as inefficient isolation, high heterogeneity, low yield, and biological variability, which limit large-scale production and standardized application. Addressing these issues can further enhance the clinical application potential of EVs and reduce their limitations in drug delivery."

Reviewer 2 Report
Comments and Suggestions for Authors
In the manuscript by Zhang H. et al., authors made a literature review of the role of extracellular vesicles (EVs) in the settlement of cancer drug resistance, underlaying also the possible role of EVs as drug delivery tools and/or as negative modulators of drug resistance. In the first part of the review authors introduced data on drug resistance in cancer and on EVs, then they illustrated the mechanisms through EVs are able to mediate drug resistance in both donor and acceptor cells. In the second part of the review authors analyzed the strategies used to counteract EV-mediated drug resistance through EVs secretion inhibition and/or by engineering EVs as drug as well as anticancer ncRNAs transporters.
The MS by Zhang H. et al. is well conceived and well organized and the topic is extensively illustrated. I do not find any major issue even I suggest the authors to revise the following sentence (cfr. Introduction line 13):
“These vesicles can be identified and separated based on different molecular markers.”
Written that way this sentence is misleading because it seems that there are specific markers that allow the identification and possibly the purification of specific EV subset (i.e. exosomes and microvesicles). This is not the case, indeed recommendation from MISEV (Minimal information for studies of extracellular vesicles) suggests the use of the terms small EV and large EV rather than exosomes and microvesicles due to the overlap in size of these two subset and the lack of uniquely expressed markers.
Authors should report these issues, referring to these articles: Welsh JA et al. 2024 DOI: 10.1002/jev2.12404; Witwer KW et al. 2021 DOI: 10.1002/jev2.12182; Witwer KW et al. 2017 DOI: 10.1080/20013078.2017.1396823.
Otherwise the manuscript is suitable for publication on Molecules.
Author Response
Reviewer 2
In the manuscript by Zhang H. et al., authors made a literature review of the role of extracellular vesicles (EVs) in the settlement of cancer drug resistance, underlaying also the possible role of EVs as drug delivery tools and/or as negative modulators of drug resistance. In the first part of the review authors introduced data on drug resistance in cancer and on EVs, then they illustrated the mechanisms through EVs are able to mediate drug resistance in both donor and acceptor cells. In the second part of the review authors analyzed the strategies used to counteract EV-mediated drug resistance through EVs secretion inhibition and/or by engineering EVs as drug as well as anticancer ncRNAs transporters.
The MS by Zhang H. et al. is well conceived and well organized and the topic is extensively illustrated. I do not find any major issue even I suggest the authors to revise the following sentence (cfr. Introduction line 13): "These vesicles can be identified and separated based on different molecular markers."
Written that way this sentence is misleading because it seems that there are specific markers that allow the identification and possibly the purification of specific EV subset (i.e. exosomes and microvesicles). This is not the case, indeed recommendation from MISEV (Minimal information for studies of extracellular vesicles) suggests the use of the terms small EV and large EV rather than exosomes and microvesicles due to the overlap in size of these two subset and the lack of uniquely expressed markers.
Authors should report these issues, referring to these articles: Welsh JA et al. 2024 DOI: 10.1002/jev2.12404; Witwer KW et al. 2021 DOI: 10.1002/jev2.12182; Witwer KW et al. 2017 DOI: 10.1080/20013078.2017.1396823.
Otherwise the manuscript is suitable for publication on Molecules.
Response: Many thank for your insightful reminder. After carefully reviewing the literature on MISEV, we found that this is indeed the case. The separation of EVs subset is indeed not classified based on unique markers. Therefore, we deleted this sentence.

Reviewer 3 Report
Comments and Suggestions for Authors
An interesting topic of the review paper, logical construction of the manuscript, selected literature published in the last 10 years, with a significant preponderance of those published in the last 5 years, written in good English.
Unfortunately, the fundamental problem is the lack of scientific integrity. The authors, when presenting particular issues on the current state of knowledge on the role of EVs in drug resistance, cite articles whose content lacks the information referenced by the authors. Since this happened repeatedly, the examples below, I have concluded my evaluation of this manuscript at the first paragraph stage in Chapter Two.
Chapter 1. Introduction
1. Incorrectly selected references. The content of these articles lacks the information referenced by the authors in the following sentences:
- Drugs are crucial and highly effective in controlling cancer, ultimately offering unparalleled anticancer benefits [2].
- …. which leads to resistance to multiple cancer drugs [6].
2. Incorrectly stated microvesicles size range (page1). Should be 100-1000 nm instead of 50-500 nm.
Chapter 2. Mechanisms by which EVs regulate drug resistance in donor cells
3. The study described in reference #8 did not involve patients with prostate cancer (page 2). Besides, it is difficult to say that the 1-fold increase is a real increase. The term “1-2-fold increase” appears twice on page 2.
4. The Authors stated that “Moreover, temozolomide-treated glioblastoma cells exhibited a 1-2 fold increase in EVs count [12, 13] – page 2”. Meanwhile, a 3-fold increase is reported in both references #12 and# 1 on page 13 and page 245,respectively.
5. The Authors stated that “Docetaxel induced triple-negative breast cancer cells to secrete a greater number of small EVs (sEVs) with a diameter not exceeding 500 nm [14]. – page 2”. The authors consider such vesicles not exceeding 500 nm in diameter to be sEVs. Meanwhile, minimal information for studies of extracellular vesicles (MISEV2023) suggest operational use of terms “small” and “large EVs” when characterizing isolated samples, with 200 nm being the discriminating diameter value. Moreover, in the cited reference #14, the isolated EVs were generally less than 200 nm in diameter, as is clearly shown in Figure 1.
6. The Authors stated that “In vitro experiments with NSCLC and chronic myeloid leukemia cells demonstrated that the number of EVs released by drug-resistant cancer cells surpassed those released by drug-sensitive counterparts [15] – page 2”. In the cited reference #15, I found no confirmation that suppression actually occurs. The abstract of reference #15 summarized the results obtained this way (quote): Our results showed that MDR cells released more EV than their drug-sensitive counterparts, and that drug-sensitive cells captured more EV than their MDR counterparts. This difference in the release and uptake of EV may be related to differences in the endocytic pathway between drug-sensitive and MDR cells. Importantly, manipulation of the recycling pathway affected the response of drug-sensitive cells to doxorubicin cells to doxorubicin treatment.
7. The Authors stated that” However, conflicting evidence suggests that EVs released by drug-resistant and drug-sensitive cancer cells exhibit no significant difference in count [16] – page 2”. In the cited reference #16 I did not find in section 2.2 Cell culture any information that drug-sensitive and drug-resistant prostate cancer cells were used for the studies described in this article.
8. The Authors stated that “The change of EVs count may be a fundamental characteristic and predictor of resistance [15, 17, 18]. – page 2”. In the cited references #15, #17 and # 18, I found no confirmation that the amount of EVs released could be taken as a predictor of resistance.
9. The Authors stated that: “In addition, future studies should characterize distinct EVs subgroups released by cancer cells under various cancer drug treatments and consider utilizing the EVs count as a monitoring method for cancer drug therapy responses [19]- page 2“. Meanwile, the cited article (#eference #19) focuses on (quote): In this review, we discuss recent advances in the identification of immune checkpoints using EVs and CTCs, their potential applications as predictive biomarkers for ICI therapy, and their prospective use as innovative clinical tools, considering that CTCs have already been approved by the Food and Drug Administration (FDA) for clinical use, but providing good reasons to intensify the research on both.
Author Response
Reviewer 3
An interesting topic of the review paper, logical construction of the manuscript, selected literature published in the last 10 years, with a significant preponderance of those published in the last 5 years, written in good English.
Unfortunately, the fundamental problem is the lack of scientific integrity. The authors, when presenting particular issues on the current state of knowledge on the role of EVs in drug resistance, cite articles whose content lacks the information referenced by the authors. Since this happened repeatedly, the examples below, I have concluded my evaluation of this manuscript at the first paragraph stage in Chapter Two.
Chapter 1. Introduction
- Incorrectly selected references. The content of these articles lacks the information referenced by the authors in the following sentences:
- Drugs are crucial and highly effective in controlling cancer, ultimately offering unparalleled anticancer benefits [2].
Response: Thank you very much for your kind reminder. We agree that the reference selection here is inaccurate. To resolve this problem, we deleted the original content and changed it to: "Drug therapies have provided sustained benefits for cancer control." The original reference [2] has been deleted.
- which leads to resistance to multiple cancer drugs [6].
Response: Thank you for your detailed review. We have replaced the reference here with an accurate one. The new reference is: "5. Bucci-Muñoz M, Gola AM, Rigalli JP, Ceballos MP, Ruiz ML. Extracellular Vesicles and Cancer Multidrug Resistance: Undesirable Intercellular Messengers?. Life (Basel). 2023;13(8):1633. Published 2023 Jul 27. doi:10.3390/life13081633".
- Incorrectly stated microvesicles size range (page1). Should be 100-1000 nm instead of 50-500 nm.
Response: Thank you for your kind reminder. We uniformly classify EVs according to the description of MISEV. According to MISEV, we divide EVs into large EVs and small EVs, and define exosomes as a subgroup of small EVs. Microvesicles are defined as a subtype of EVs. The contents have been changed to: "EVs are generally classified into large EVs (diameter >200 nm) and small EVs (sEVs,diameter <200 nm). Exosomes, a subset of sEVs, originate from the endosomal system and are released through the multivesicular body (MVB) pathway, whereas microvesicles, derived from the plasma membrane, typically refer to larger EVs or all EVs [6, 7]."
Chapter 2. Mechanisms by which EVs regulate drug resistance in donor cells
- The study described in reference #8 did not involve patients with prostate cancer (page 2). Besides, it is difficult to say that the 1-fold increase is a real increase. The term "1-2-fold increase" appears twice on page 2.
Response: Thank you for your kind reminder. The content here is wrong. We have carefully checked the manuscript again and found that the reference [8] does not describe patients, but prostate cancer cells. So we changed the relevant content.
The content before the modification is: "One study reported that after 24 hours of doxorubicin treatment, EVs secretion increased 1-2 fold in prostate cancer patients, confirmed both in vitro and in the blood of experimental mice."
The modified content is: "After 24 hours of exposure to doxorubicin, EVs in the plasma of tumor-bearing mouse were significantly increased compared to controls." (The original conclusion is shown in Figure 5C of section PDT and chemotherapy induced vesicle release in vivo in the reference [8].)
- The Authors stated that "Moreover, temozolomide-treated glioblastoma cells exhibited a 1-2 fold increase in EVs count [12, 13] – page 2". Meanwhile, a 3-fold increase is reported in both references #12 and# 1 on page 13 and page 245,respectively.
Response: Thank you for your insightful reminder. Considering the statistical test results in the reference, we have revised the description to make the content more accurate.
The original content is: "Moreover, temozolomide-treated glioblastoma cells exhibited a 1-2 fold increase in EVs count."
The revised content is: "Moreover, temozolomide-treated glioblastoma cells exhibited significant increase in EVs count [12, 13]." (For the original results, see Figure 1C in reference [12] and Figure 5B in reference [13].)
- The Authors stated that "Docetaxel induced triple-negative breast cancer cells to secrete a greater number of small EVs (sEVs) with a diameter not exceeding 500 nm [14]. – page 2". The authors consider such vesicles not exceeding 500 nm in diameter to be sEVs. Meanwhile, minimal information for studies of extracellular vesicles (MISEV2023) suggest operational use of terms "small" and "large EVs" when characterizing isolated samples, with 200 nm being the discriminating diameter value. Moreover, in the cited reference #14, the isolated EVs were generally less than 200 nm in diameter, as is clearly shown in Figure 1.
Response: Thank you for your reminder. We have changed the content according to MISEV.
The original content is: "Docetaxel induced triple-negative breast cancer cells to secrete a greater number of small EVs (sEVs) with a diameter not exceeding 500 nm."
The modified content is: "Docetaxel induced triple-negative breast cancer cells to secrete a greater number of sEVs [14]."
- The Authors stated that "In vitro experiments with NSCLC and chronic myeloid leukemia cells demonstrated that the number of EVs released by drug-resistant cancer cells surpassed those released by drug-sensitive counterparts [15] – page 2". In the cited reference #15, I found no confirmation that suppression actually occurs. The abstract of reference #15 summarized the results obtained this way (quote): Our results showed that MDR cells released more EV than their drug-sensitive counterparts, and that drug-sensitive cells captured more EV than their MDR counterparts. This difference in the release and uptake of EV may be related to differences in the endocytic pathway between drug-sensitive and MDR cells. Importantly, manipulation of the recycling pathway affected the response of drug-sensitive cells to doxorubicin cells to doxorubicin treatment.
Response: The content here may cause ambiguity. We have changed it.
The original content is: "In vitro experiments with NSCLC and chronic myeloid leukemia cells demonstrated that the number of EVs released by drug-resistant cancer cells surpassed those released by drug-sensitive counterparts [15]."
The changed content is: "In vitro experiments with NSCLC and chronic myeloid leukemia cells demonstrated that the number of EVs released by drug-resistant cancer cells was significantly greater than those released by drug-sensitive counterparts [15]."
- The Authors stated that" However, conflicting evidence suggests that EVs released by drug-resistant and drug-sensitive cancer cells exhibit no significant difference in count [16] – page 2". In the cited reference #16 I did not find in section 2.2 Cell culture any information that drug-sensitive and drug-resistant prostate cancer cells were used for the studies described in this article.
Response: Thank you for your kind reminder. There is a mistake in the details here. The reference article reported that androgen treatment affected the number and cargo of sEVs in prostate cancer patients, but there was indeed no resistance-related details. Therefore, we corrected the content to make it accurate.
The original content is:"conflicting evidence suggests that EVs released by drug-resistant and drug-sensitive cancer cells exhibit no significant difference in count".
The changed content is: "clinical studies have reported that androgen therapy affects the count and cargo of sEVs in prostate cancer patients [16], but there is currently no definitive evidence linking clinical androgen resistance to blood sEVs."
- The Authors stated that "The change of EVs count may be a fundamental characteristic and predictor of resistance [15, 17, 18]. – page 2". In the cited references #15, #17 and # 18, I found no confirmation that the amount of EVs released could be taken as a predictor of resistance.
Response: This is indeed over-speculation.
We changed the content to "The change of EVs count may be a fundamental characteristic of resistance [15, 17, 18]."
- The Authors stated that: "In addition, future studies should characterize distinct EVs subgroups released by cancer cells under various cancer drug treatments and consider utilizing the EVs count as a monitoring method for cancer drug therapy responses [19]- page 2". Meanwile, the cited article (#eference #19) focuses on (quote): In this review, we discuss recent advances in the identification of immune checkpoints using EVs and CTCs, their potential applications as predictive biomarkers for ICI therapy, and their prospective use as innovative clinical tools, considering that CTCs have already been approved by the Food and Drug Administration (FDA) for clinical use, but providing good reasons to intensify the research on both.
Response: Thank you for your reminder. We have replaced the reference with a more accurate one.
The replaced reference is: "19. Zhou E, Li Y, Wu F, et al. Circulating extracellular vesicles are effective biomarkers for predicting response to cancer therapy. EBioMedicine. 2021;67:103365. doi:10.1016/j.ebiom.2021.103365". This reference summarized the potential of circulating EVs as biomarkers for predicting cancer patients' responses to various cancer therapies. The results suggest that circulating EVs hold promise as effective biomarkers for predicting responses to cancer treatment.
In addition, we checked the following content and checked the description of MISEA and the problem of changing fold.
